# Integrating Human Waste with Microbial Fuel Cells to Elevate the Production of Bioelectricity

**DOI:** 10.3390/biotech11030036

**Published:** 2022-08-22

**Authors:** Chetan Pandit, Bhim Sen Thapa, Bhagyashree Srivastava, Abhilasha Singh Mathuriya, Umair-Ali Toor, Manu Pant, Soumya Pandit, Deepak-A. Jadhav

**Affiliations:** 1School of Basic Science and Research, Sharda University, Greater Noida 201306, India; 2Department of Biological Sciences, WEHR Life Sciences, Marquette University, Milwaukee, WI 53233, USA; 3Department of Bioscience and Biotechnology, Banasthali Vidyapith, Jaipur 304022, India; 4Ministry of Environment, Forest and Climate Change, New Delhi 110003, India; 5Institute of Animal Life Science, Kangwon National University, Chuncheon 24341, Korea; 6Department of Life Sciences, Graphic Era Deemed to Be University, Dehradun 248002, India; 7Department of Environmental Engineering, Korea Maritime and Ocean University, 727 Taejong-ro, Yeongdo-gu, Busan 49112, Korea

**Keywords:** human waste, human urine, microbial fuel cell, bioelectricity production, waste management

## Abstract

Due to the continuous depletion of natural resources currently used for electricity generation, it is imperative to develop alternative energy sources. Human waste is nowadays being explored as an efficient source to produce bio-energy. Human waste is renewable and can be used as a source for an uninterrupted energy supply in bioelectricity or biofuel. Annually, human waste such as urine is produced in trillions of liters globally. Hence, utilizing the waste to produce bioenergy is bio-economically suitable and ecologically balanced. Microbial fuel cells (MFCs) play a crucial role in providing an effective mode of bioelectricity production by implementing the role of transducers. MFCs convert organic matter into energy using bio-electro-oxidation of material to produce electricity. Over the years, MFCs have been explored prominently in various fields to find a backup for providing bioenergy and biofuel. MFCs involve the role of exoelectrogens which work as transducers to convert the material into electricity by catalyzing redox reactions. This review paper demonstrates how human waste is useful for producing electricity and how this innovation would be beneficial in the long term, considering the current scenario of increasing demand for the supply of products and shortages of natural resources used to produce biofuel and bioelectricity.

## 1. Introduction

Sanitation methods in any location determine the quality of human life and its development index. Most rural populations in developing nations lack basic sanitation services, and a major segment of the population lacks access to sanitary living conditions. According to an earlier sanitation assessment, developing nations will confront serious sanitation challenges in the next decades. Rapidly growing countries lack enough onsite sanitation facilities for around 40–50 percent of their overall population [1]. Despite various long-standing international programs and complementary methods targeted at tackling inadequate sanitation in vulnerable nations, the numbers of people living without proper sanitation facilities, as well as the related health risks and impacts, remain alarming. Currently, 2.3 billion people lack access to proper sanitation services, with 494 million resorting to open defecation [2] (Figure 1). Currently, several forms of site sanitation procedures are used, which are poorly managed and pollute water bodies. Several new initiatives have lately been started in developing and developed nations to build toilets and eliminate open defecation to improve public hygiene [3].

Researchers are focusing on the development of waste-to-energy recovery systems in order to use wastewater as a possible renewable energy resource. One such waste-to-energy recovery system is the MFC, which transfers energy contained in organics from the waste substrate into electrical output by using bacteria as a biocatalyst in different electrochemical redox processes [4,5,6,7,8,9]. This biofuel cell can treat effluents ranging from monomer sugars (i.e., acetate, glucose) to composite industrial wastewater and lignocellulosic effluent [10,11]. However, a prospective source, i.e., human feces, has not been widely used in MFC. In addition to treating complicated industrial wastewater, there are limited publications on treating human excreta and urine in MFC [12,13]. Cow dung, cow urine, swine wastewater, elephant pee, and simulated human wastes have previously been efficiently handled in MFCs [14]. Human excreta and urine include large levels of carbon and nutrients, similarly to animal waste; however, pharmaceutical leftover chemicals and pathogenic loads are key additional concerns in human waste, increasing the complexity of human waste treatment. Human waste includes a substantial amount of organic and nutritional molecules (ammonium-nitrogen found in urine) when measured in terms of energy value. Human urine contains around 2% urea by weight. As opposed to protons in water, four hydrogen atoms per urea molecule are chemically bonded in a loose framework to the molecule. Breaking such chemical bonds requires less energy and produces more protons and electrons in electrochemical processes [14]. On the other hand, human feces have a significant concentration of carbonaceous chemicals, which can liberate electrons during microbial oxidation.

These electrons are gathered in the MFC’s external electrical (load-bearing) circuit to provide direct current (Figure 2). Even septic tank sludge may be efficiently hydrolyzed and used as a feed material for biocatalytic oxidation in MFCs [15]. Thus, during the projection of sanitation from ‘better’ to ‘sustainable sanitation’ in the circular economy using this bio-electrochemical technology, human waste must not be seen as dangerous waste but as a valuable resource that may be reutilized [16]. Implementing MFCs in new or existing septic tank systems is a possible alternative for enhanced sanitation facilities. Given the simple substrate with practically continuous flow, using MFCs for human waste treatment can aid in environmental and improved sanitary facilities, particularly in rural regions, while producing power. MFCs have already been tested at pilot size with a synthetic acetate-based substrate (85 L MFC) [17] and genuine municipal wastewater (255 L MFC) [18].

Urine can be used as a feedstock for the MFC, and You et al. (2021) have demonstrated this in their study with a production of 3.91 ± 0.27 mW, 38.9 ± 1.1% COD elimination rate and 15.1 ± 3.4% coulombic efficiency [19]. Urban fecal effluent underwent biochemical treatment, and the processed wastewater was used to make fertilizers by Hu et al. (2020). According to the findings, liquid organic fertilizer can extend the growing season of Chinese orchids by 3–4 days and okra by 12.89% when compared to untreated urine [20]. In a low-cost MFC, Sabin et al. (2022) employed pre-treated human urine that had been deprived of phosphate, nitrogen, and ammonium. The performance of MFCs using hydrolyzed actual urine and synthetic urine as feedstocks was compared to the performance of MFCs using waste bottoms as feedstock. After 32 days of operation, MFCs with waste bottoms generated 16.2 ± 14.8 mW m_cat_^−2^ (2.14 ± 1.95 W m_Cat_^−3^), which is comparable to 93% of the mean power density attained by hydrolyzed urine. Over the whole testing period, waste bottoms had a coulombic efficiency that was 32.3 ± 4.1% greater than that of urine. Waste bottoms prevented the urea hydrolysis and phosphate precipitation from urine from fouling the ceramic membrane separator [21].

Fangzhou et al. (2011) used wastewater containing large amounts of organic compounds, including human waste, as the substrate for an MFC to produce energy, and factors affecting that capability were investigated. When a two-chamber MFC was fed with the actual human feces wastewater and operated for 190 h, the removal efficiency of total chemical oxygen demand (TCOD), soluble chemical oxygen demand (SCOD), and NH^4+^ achieved 71%, 88%, and 44%, respectively. Additionally, the highest power density was 70.8 mW/m^2^ [22]. Kretzschmar et al. (2016) used MFC based on cardboard electrodes for human feces treatment. Cardboard electrodes help in developing a low-cost MFC. The MFC yields power output of 15.09 ± 5.18 μA cm^−2^ [23]. In Ghana, Castro et al. (2014) created an MFC that was used to handle organic matter removal. This investigation indicated that power generation was 3.40 ± 0.01 mW/m^2^ and nitrogen removal was 76.8 ± 7.1% [24].

About 50–60% of organic matter and nutrients are directly dumped into water streams through traditional site sanitation procedures, losing the energy value of human feces in the process. High levels of wastewater treatment efficiency and the facilitation of resource recovery possibilities cannot be handled by existing septic tanks. Because of this, septic tank systems must be engineered to capture energy from human waste while also enhancing the removal of organic matter by merging them with cutting-edge advanced treatment systems, such as microbial fuel cells (MFCs). The current review concentrated on the use of MFCs in conjunction with septic tanks for the on-site treatment of human waste in this direction. Additionally, the most recent developments in this research and the difficulties this system encountered when implemented at a field scale were explored.

## 2. Changes over Time in Human Waste Treatment

### 2.1. Conventional Methods

Improper waste management causes environmental degradation, unpleasant odors, the development and reproduction of insects and rodents, and the transfer of waterborne illnesses such as typhoid and cholera. In rural areas, several forms of in situ sanitation solutions, such as aqua privy and soak pit privy, borehole latrines, dug well latrines, conventional septic tank (CST) with soak pit arrangement, composting toilets, and so on, are now preferred [25]. If not effectively managed, these methods are to blame for water pollution. Apart from different decentralized sanitation systems, the traditional septic tank system with a soak pit layout is most commonly employed in urban and rural areas since it allows sludge to settle for a longer period in the sedimentation pit [26,27]. A septic tank, in other terms, is a sedimentation and digesting tank for these settled solids. The septic tank can only remove around 30–40% of organic matter from sewage through anaerobic digestion, and any residual organics or pathogens are disposed of in the environment without treatment [28,29]. To achieve better treatment efficiency, it is essential to redesign the traditional septic tank system more effectively and efficiently, with extra measures to handle the difficulties mentioned above. Figure 3 shows conventional techniques for human waste treatment.

### 2.2. Modern Methods

Water, plant nutrients, organic matter, and energy are all accessible in wastewater and human excreta. Sanitation systems intended for the safe and efficient recovery of resources can play a significant part in the overall resource management of a community [30,31]. Recovering the resources inherent in excreta and wastewater (such as nutrients, water, and energy) aids in the achievement of sustainable development goal 6 (SDG6) and other SDGs. Ref. [32] Combining wastewater and human excreta with other organic waste such as manure, food, and crop waste for resource recovery can be efficient (Figure 4). Excreta is most commonly reused in agriculture as fertilizer and soil conditioner. For sanitation with agriculture, this is often known as a “closing the loop” strategy [33,34]. It is a key component of the ecological sanitation strategy. The shape of the excreta being reused determines the reuse options: it might be excreta on its own, mixed with some water (fecal sludge), or excreta mixed with large volumes of water (domestic wastewater or sewage).

#### 2.2.1. Fertilizers

Human excreta is an undiscovered fertilizer resource. For example, the potential quantities of nutrients that may be retrieved from human excreta are similar to the total fertilizer consumption in continental Africa [35]. As a result, reuse can help enhance food output while providing an alternative to chemical fertilizers, typically out of reach for small-holder farmers. However, the nutritional content of human excreta is mostly determined by food intake. Mining activities provide mineral fertilizers, which may contain heavy metals [36]. Heavy metals such as cadmium and uranium are found in phosphate ores and can enter the food chain via mineral phosphate fertilizer. This does not apply to excreta-based fertilizers (unless the human diet was already tainted beyond permissible levels), which is a benefit [37]. Organic fertilizers’ fertilizing ingredients are typically bonded in carbonaceous reduced compounds. When partially oxidized, as in compost, the fertilizing minerals are adsorbed on the breakdown products (humic acids, for example) [38]. As a result, they have a slower release impact and are often less rapidly leached than mineral fertilizers.

#### 2.2.2. Pathogen Removal

Temperature is a treatment parameter linked to pathogen inactivation across all pathogen groups: Most infections can be inactivated by temperatures exceeding 50 °C. As a result, heat sanitization is used in various methods, including thermophilic composting, thermophilic anaerobic digestion, and even solar drying [39]. Alkaline circumstances can also deactivate pathogens (pH more than 10). This can be accomplished by the use of ammonia sanitization or lime treatment.

#### 2.2.3. Materials

Human urine is a universally underused organic resource disposed of by the human body. We report the first “proof of concept” of a simple, potentially cost-effective, and novel template-free approach to synthesizing highly porous carbon, including heteroatoms such as N, S, Si, and P, using human urine waste as a single precursor for carbon and multiple heteroatoms. High porosity is achieved by removing inherently present salt particles from as-prepared “Urine Carbon” (URC), and multiple heteroatoms are naturally doped into the carbon, eliminating the need for troublesome, expensive pore-generating templates as well as extra expensive heteroatom-containing organic precursors. Furthermore, the isolation of rock salts is an added benefit of the current effort [40]. The technique is simple but effective, producing naturally doped conductive hierarchical porous URC with superior electrocatalytic ORR activity comparable to state-of-the-art Pt/C catalysts and significantly improved durability and methanol tolerance, demonstrating that the URC can be a promising alternative to expensive Pt-based electrocatalysts for oxygen reduction reaction (ORR) [41]. The ORR activity of the carbon framework may be addressed in terms of heteroatom doping, surface characteristics, and electrical conductivity. Various techniques for human waste management by modern techniques are listed in Table 1.

## 3. MFC Technology

The global use of energy has been on an upward trajectory in recent decades. Energy sources are classified as fossil fuels, renewable sources, and nuclear sources, with non-renewable forms of energy, which account for a significant portion of total energy consumption, classified as nuclear and fossil energy. However, several countries worldwide have made remarkable efforts to find a realistic solution to the energy crisis by focusing on renewable energy sources such as solar energy, wind energy, and water energy. As a result of this strategy, one of the most recently proposed alternative energy sources is the fuel cell (FC), which generates energy using high-value metal catalysts. Microbial fuel cells (MFCs) are a type of FC that generates bioelectricity by employing an active microbe as a biocatalyst in an anaerobic anode compartment. Almost all MFCs, as shown in Figure 5, are built up of anode and cathode chambers that are physically separated by a proton exchange membrane (PEM). The active biocatalyst in the anode oxidizes the organic substrates, creating electrons and protons. The PEM transports the protons to the cathode chamber, whereas the external circuit transports the electrons. Protons and electrons react in the cathode chamber as oxygen is converted to water. It is worth mentioning that the active biocatalyst in the anode compartment oxidizes the carbon sources or substrates, creating electrons and protons. Equation (1) shows the anodic reaction of acetic acid as another fascinating example. Because oxygen in the anode chamber affects electricity generation, a viable system for keeping bacteria away from oxygen must be devised (anaerobic chamber for anodic reaction).
C_2_H_4_O_2_ + 2H_2_O → 2CO_2_ + 8e^−^ + 8H^+^(1)
2O_2_ + 8H^+^ + 8e^−^ → 4H_2_O(2)

A biocatalyst can be separated from oxygen by sandwiching a membrane between two chambers that allow charge to be transmitted between the electrodes, the anode chamber, where bacteria grow, and the cathode chamber, where electrons react with oxygen. MFC technology has come a long way in the last few decades. However, it has encountered several challenges in terms of scale-up and practical application, including turbulence in each compartment and membrane resistance in the proton transportation process. MFCs have also experienced two bottleneck concerns in electricity generation. Power generation and substrate concentrations in MFCs exhibit direct connections, albeit varied, in each system. Power generation will be inhibited if the substrate concentration exceeds a certain threshold. MFC output is restricted, and high internal resistance wastes most of the power generated by MFCs. The proton exchange membrane (PEM), which separates the anode and cathode chambers, has been identified as the principal source of increased internal resistance (R_in_) in two MFC chambers.

## 4. Factors Affecting the Performance of an MFC

An MFC converts the chemical energy of organic elements in the substrate into electrical energy. This conversion occurs when bacteria in the anodic chamber oxidize the substrate (electron donors such as oxygen, nitrate, or sulfur species) and create electrons and protons. Several factors influence the effectiveness of MFCs and their energy output in wastewater treatment. Recognizing and understanding these elements is critical for having a highly efficient MFC. Because of their metabolism and the mediators they utilize to convey electrons to the anode, microorganisms in the anodic chamber are important. A variety of substrates may be employed as an electron donor source in the MFC and oxidized by microorganisms.

### 4.1. Electron Transfer Mechanism

Shuttles or electron mediators should convey electrons from the anodic chamber to the anode. Colors such as methylene blue, neutral red, thionine, methyl viologen, and humic acid are commonly used as mediators. Certain characteristics are required of mediators, such as the ability to cross through the bacterial membrane and reach the reductive species within the bacterium, where they are reduced during microbial metabolism [42]. The redox potential of the mediator should be equivalent to that of the reductive metabolite. The mediator must not interact with any of the bacteria’s other metabolites. Reduced mediators should be able to travel easily out of the cell and to the anode, where they may be oxidized [43]. At the electrode surface, the electrochemical kinetics of the mediator-reduced state oxidation process must be fast.

### 4.2. Microbial Metabolism and Cell Potential

The metabolic pathway of microorganisms and the consequent potential of the anode are the most important parameters in determining cell potential [44]. Bacterial catabolism is the rate-limiting step in MFCs. Heterotrophic organisms obtain energy from the oxidation of organic molecules [45]. Because of the presence of exogenous oxidants, or external terminal electron acceptors, the anodic chamber undergoes two basic metabolic processes: the respiratory chain and fermentation [46,47]. The substrate is oxidized throughout the respiratory chain, releasing electrons that move through a redox cascade to an externally accessible terminal electron acceptor [48]. The higher the positive redox potential of a terminal electron acceptor with a particular substrate, i.e., electron donor, the greater the energy gain for an organism.

### 4.3. Substrate

Among the several substrates, wastewater is a resource-rich medium that MFCs may handle. Several reports have been published on the direct generation of electricity from complex organic wastewater, including municipal, swine wastewater, dairy wastewater, slaughterhouse wastewater, rice mill wastewater, tannery wastewater, cassava mill wastewater, molasses wastewater, refinery wastewater, brewery wastewater, winery wastewater, chemical wastewater, sulfide-rich wastewater, landfill leachate, food waste leachate, azo dye, and solid substrates such as rice straw [49,50,51,52]. MFCs can also simultaneously remove sulfide and nitrate from synthetic wastewater and sulfate–sulfide-rich wastewaters [53].

### 4.4. Anode

The anode material’s main properties are conductivity, stability, biocompatibility, non-corrosiveness, and surface area. Furthermore, the construction process of the electrodes and the architecture of the MFC have an impact on MFC performance. Despite its high conductivity, copper is not a useful anode material owing to its bacterial toxicity. Although carbon is a better electrode material for bacterial adhesion, it has a low conductivity for transferring electrons. It is available in carbon felt, fabric, foam, paper, and fiber forms. Copper anodes generate less electricity (2 mW/m^2^) than stainless steel (12 mW/m^2^) or carbon cloth (880 mW/m^2^) anodes. This is due to the CC’s larger specific surface area compared to the metal mesh [54]. Graphite rods or plates are more adaptable because they are easier to handle, less costly, and have a fixed surface area [55]. When compared to alternative electrode materials for usage in MFCs, CCs have a higher mechanical strength and porosity. However, they have certain drawbacks, including a high cost that renders them uneconomical. Because of their larger specific surface area, more active microbe–electrode–electrolyte interaction, and higher electron transfer efficiency, graphene-based electrodes are a promising material that has recently received attention [56].

### 4.5. Cathode

The materials and procedures used to manufacture the cathode electrode can also have an impact on the performance of MFCs. Platinum electrodes are the most often utilized oxygen reduction catalysts. They outperform graphite and carbon electrodes; however, they are more costly and unsustainable due to toxic chemicals in microbial solutions in open-air biocathodes [57]. Because Pt is expensive for MFC construction, decreasing or eliminating the demand for Pt can minimize the MFC’s capital cost. In certain research, commercially made cathodes with a Pt loading of 0.5 mg/cm^2^ were employed [58]. It has been established that a Pt loading of 0.1 mg/cm^2^ on the water side of an air cathode does not affect power density. Santoro et al. discovered that during the starting phase, Pt-based cathodes produced more power (330 mW/m^2^) than Pt-free cathodes (253 mW/m^2^), but the difference evaporated after a few weeks of operation [59]. Carbon materials, nitrogen functionalized carbon nanotubes (CNTs), mesoporous nitrogen-rich carbon, and cotetramethyl phenylporphyrin can all be used in place of Pt.

### 4.6. Operating Conditions

Temperature, pH, ionic strength, and salinity are the most important characteristics of bacterial growth, and they all have an impact on MFC performance. MFCs generally have benign reaction conditions, such as ambient temperature, normal pressure, and neutral pH.

## 5. Integrating Human Waste with Microbial Fuel Cells

Several scientific and technological advancements during the last 100 years of study have led to the advancement of MFC technology toward commercialization. Given the current state of MFC research, field deployment of this technology is limited by its low power output and related manufacturing costs. Field use of MFC may be an appropriate and realistic solution for long-term sanitation to overcome the poor treatment effectiveness of the currently used CST system. Furthermore, human waste (excreta and urine) can be regarded as a sustainable golden gold (substrate) for use in MFC to collect energy during successful wastewater treatment while also recovering by-products (Figure 6) and facilitating the reuse of treated water. From sustainable sanitation, environmental concerns, and public health, such an eco-friendly bio-electrochemical system is presented as an independent and renewable energy source. As with sediment MFCs, field applications of MFCs for sanitation are only viable and practicable if septic tank systems are adapted to modern bio-electrochemical systems. As a result, this newly discovered bioelectrochemical technology may become a viable alternative for efficient and sustainable sanitation and bio-energy recovery. Such bioelectric toilet (BET) technology can provide effective organic matter removal, nutrient removal, sulfur removal, effluent disinfection, efficient fecal sludge degradation, odor removal, and valuable resource recovery while also generating power to power LED bulbs, biosensors and other electronic appliances. The power density of different substrates can be seen in Table 2.

Disposing of human excreta by MFCs may cleanse toilet waste and recover electrical power [65]. Kretzschmar et al. constructed an eLatrine MFC with low-cost corrugated cardboard as an electrode material and human feces as a substrate during anodic oxidation to capture potential energy from human waste [23]. Perlow’s first investigation on MFC waste treatment by microbial oxidation produced a low open-circuit voltage of 200 mV [66]. In terms of pollutant degradation, a dual-chamber MFC handling genuine human waste removed 71 percent of total chemical oxygen demand (TCOD), 88 percent of soluble COD, and 44 percent of NH_4_^+^ while producing 70.8 mW/m^2^ of electricity [22]. When three units of air cathode column MFCs were coupled in a stack configuration during wastewater treatment at a septic tank, a power density of 142 ± 6.71 mW/m^2^ was obtained [67]. Leton et al. recently used septage wastewater in a 20 L MFC and obtained 99.8 percent, 86.4 percent, and 82.8 percent total suspended solids (TSS), biochemical oxygen demand (BOD), and nitrate removal efficiencies when operating at 1000 ohms [68]. In a plexiglass MFC, bacterial sp. *Pseudomonas otitidis* AATB4 isolated from septic tank effluent demonstrated a current density of 800 mA/m^2^ and coulombic efficiency (CE) of 15% [69]. Thus, researchers have made limited attempts to collect energy from human waste, but lower power output hinders the scaling up of such MFCs [24].

Fecal sludge is a semisolid slurry (either undigested or partially digested) and biomass produced during the collection, storage, or treatment of human excreta, as well as blackwater, which takes longer to degrade than human excreta [70,71]. Using MFC technology, chemical energy held in fecal sludge or activated sludge organic matter may be bio-electrochemically transformed into electrical energy. During the experiment, an operating voltage of 0.45–0.65 V was reached with a CE of 1.5–4.3 percent in a single-chambered MFC using fecal matter as the substrate [72]. Furthermore, the microbial population in fecal sludge may be employed as electrogenic inoculum, eliminating the need for external seeding during MFC startup [73].

By exerting an external potential, the chemical energy available in urine can be recovered as electricity in MFC or as hydrogen gas during electrolysis in microbial electrolysis cells (MECs) [74,75]. MFC/MECs can provide a new approach to urine treatment due to recent advancements in microbial electrochemical systems [55]. Such a urine waste-to-energy strategy for bioelectricity production in MFCs has been published recently [10], and research for practical applications to charge mobile phone batteries utilizing stacked MFC units is ongoing [75]. Zang et al. described a unique method of struvite (MgNH_4_PO_4_H_2_O) recovery and energy production from human urine utilizing a three-stage integrated MFC system [76]. Kuntke et al. used actual and synthetic urine in an MFC to accomplish ammonium recovery and concomitant energy recovery [74].

Furthermore, Ieropoulos et al. evaluated the artificial urine in a 1 mL MFC, yielding a net power production of 962.94 mW/m^3^ [75]. In 2016, Ieropoulos and colleagues [77] conducted field testing of a Pee power urinal and reached a maximum power of 800 mW when the light was linked directly to about 432 MFC units, as well as a COD removal efficiency of more than 95%. This was the first field testing of an MFC system for wastewater produced by urinals, and it demonstrated the applicability of MFCs for simultaneous power harvesting and direct pee treatment. Since then, urine has been a popular substrate in MFC for ammonia recovery [78].

## 6. Power Production

Human excreta and fecal sludge can be utilized as fuels for microbial oxidation to capture bioelectricity during treatment in a bioelectric toilet (BET) system. This mechanism can oxidize the organic materials found in human feces [79]. It is a CST and bioelectrochemical system modified with a power management system (PMS). Because CSTs lack energy collecting capacity, these extra characteristics are recommended for remote areas that do not have access to power. Advanced PMS is typically used to retain the charge generated by separate MFC units and release it in series or parallel connections, depending on the application. It is made up of a charge pump, a supercapacitor, a boost converter, electric circuitry, and a charge management circuit [80]. The charge management circuit regulates the flow of charge from stronger to weaker cells, limiting voltage reversal to some extent. The supercapacitor and booster in the PMS handle variations in charge and current caused by changes in operating conditions in the anodic chamber of the MFC. Once created, such a BET system with enhanced PMS may be utilized by individuals and communal dwellings to manage waste generated by toilets while also producing onsite power. The biggest bioelectric toilet MFC of 1.5 m^3^ capacity was created for field use at Indian Institute of Technology Kharagpur, India, for the treatment of human waste as well as the generation of power. The system comprises a hexagonal center settling chamber, five outer anodic chambers with air cathode arrangements, and one chlorinated aqueous cathodic chamber. This system comprises 49 pairs of separator electrode assemblies constructed using carbon felt as both anode and cathode electrode material and modified clayware ceramic tile with a 20% montmorillonite cation exchanger as the separator. It is one of the world’s largest MFC designs, constructed with fiber-reinforced plastic material and multiple cathode catalysts to improve performance, and it has been in continuous operation for the last 3 years. The BET is a sustainable field-size demonstration of MFC technology that treats human waste, removes organic carbon, reuses and recycles treated water, and generates energy to illuminate the toilet at night. The anoxic oxidation of ammonia to nitrogen gas and sulfide to elemental sulfur alleviates the odor problem in this system [5]. To deal with microorganisms from effluent, hypochlorite is employed as a disinfectant in the system’s final aqueous cathodic chamber [81]. This technique produces little sludge, which may be used as compost manure and fertilizer. Because its products may be exploited for economic advantage, these incentives will boost the use and spread of this MFC-based toilet waste management system. Figure 7, Figure 8, Figure 9 and Figure 10.

## 7. Techno-Economic and Cost Analysis

Recent advancements in electrode materials, the creation of novel efficient cathode catalysts, the design of MFC architecture, and the development of low-cost membranes have made MFC manufacture somewhat cost-effective. However, the success of field-scale MFC is dependent on cost economics and energy analysis from a fuel cell standpoint. Several cathode catalysts, metal-doped catalysts, and other modifications have been described before; however, these changes may not be a viable alternative for field application due to the high cost of production [84,85]. Unique, low-cost separator materials, such as ceramic, clayware, and modified cement, can be used as alternatives to expensive polymeric membranes, but their mechanical strength, electrochemical characteristics, and long-term stability must be improved. The development of low-cost, highly proton transferrable, and long-term stable separator materials is critical for the future viability of MFCs.

A dramatic increase has inspired modernization in designing new MFCs in power production and a decrease in overall capital cost investment to achieve a safe cost-benefit ratio [86]. The membrane, electrode, catalyst, and current collectors account for the majority of the cost of MFC production. Therefore, low-cost alternatives must be discovered. Trapero et al. recently completed an economic study that revealed that the MFC is a more economical and commercially feasible solution than the traditional activated sludge process, despite the pricey Pt/C catalyst employed in the cathode [87]. Furthermore, recovering energy and precious materials makes it a more sustainable solution than traditional wastewater treatment systems. In the current context, greater emphasis is being placed on making the MFC a self-sufficient and energy-positive system for wastewater treatment rather than traditional anaerobic treatment techniques [88,89]. However, various hurdles must be overcome before MFCs can realize their full potential as a multibillion-dollar worldwide market scenario.

## 8. Feasibility Studies

The benefits of MFC technology, which provides the treatment of, and energy generation from, human waste, have been discussed in the preceding sections of this study. However, various hurdles must be overcome for this technology to reach a competitive commercial level. The influence of numerous determining elements explored extensively in this study provides insight into building systems that are more efficient in meeting the aims of MFC applications in real-world settings. This study also emphasizes the importance of developing and improving economically viable, novel electrode materials with high porosity and conductivity to facilitate microbial biofilm formation on a large surface area, as well as new electrode modification techniques and the development of high current areal densities. The development of low-cost electrode materials with good bioelectrocatalytic performance is critical for the future development of MFC applications. Using suitable, biologically rich wastewater in the anode and metal-containing wastewater in the cathode, which can be supplied cheaply from various businesses, would minimize total operational and maintenance expenditures. Because not all concentrations are favorable in the beginning, especially for biocathode, there may be an external cost to maintaining the initial metal concentration in the cathode. The expense of air sparging, or external energy, tends to raise the operating costs of MFCs, which can be decreased by using a biocathode as a catalyst for oxygen reduction. Another issue is the MFC’s inefficiency in reducing low redox potential metals due to their reduced electron-accepting capacity. The specialized microbial communities serving as biocatalysts promote metal reduction/recovery in MFCs and reduce the overall cost by avoiding the need for costly catalysts and the expense of cathode aeration.

Another component that has been discovered to be critical to high MFC output is biofilm maturation. Long-term application of MFCs for metal removal and recovery must thus focus on this component with adequate biofilm acclimation and stability. Most metals beyond a certain threshold level are harmful to living cells and hence operate as one of the restrictions that affect metal removal capacity in MFCs, necessitating the employment of metal-tolerant microorganisms in the inoculum. According to certain research, cathode aeration improves metal removal by MFCs by reducing hydrogen peroxide, a powerful oxidizing agent. There is a need to search for bacteria that can establish such conditions in the cathode, eliminating the requirement for aeration and making it more cost-effective.

Given the environmental benefits of MFCs, they appear to be an eco-efficient technology that should be examined for inclusion in national policy analyses once cost-cutting measures are implemented to ensure economic viability. The use of MFCs in wastewater treatment, metal removal, and power production necessitates a greater knowledge and research effort, which will necessitate significant government investment. A detailed grasp of the decisive elements outlined above in the article is required for the commercial development of MFC technology. The sectors involved in biosensors, wastewater treatment, and education will be the primary end-users. More study is needed to reduce the external expenses associated with MFCs so that the advantages of MFCs with continuous energy generation will be economically viable. However, because MFCs are often prized for their dual advantages, insufficient power generation may impede the progress of this technology. According to predictions, North America will witness the largest increase owing to significant R&D spending. There are currently some pilot-scale applications, but the market for MFCs has yet to grow. The success of MFCs will rely heavily on integrating wastewater treatment with metal removal/recovery in energy-sustained MFC technology, which will be much more cost-effective than renewable energy production from other biomass sources.

## 9. Prospects

Septic tank wastewater can be utilized as a possible substrate in MFCs for bioelectricity production. Adopting a bioelectrochemical system in a traditional septic tank would improve its function for onsite sanitation and environmental protection. Even small changes to the existing system resulted in improved organic matter removal and increased treatment efficiency. Based on early economic calculations, Yazdi et al. indicated that a system to power an LED light of 6 W capacity every day for 4 h costs around USD 25, with great potential for improvement [67]. Although the power generated by human feces is insufficient for practical applications, with developing strategies to increase power output, energy generation from human feces will become more helpful for commercial exploration. The bio-electrochemical system presents an alternative method that looks more realistic and practicable from both a technical and economic standpoint. At present, the cost limits for incorporating microbial electrochemical technology into toilet systems are the most difficult to overcome. Compared to a traditional septic tank with a soak pit system, the land need for such toilets is far smaller. People will have simple access to an affordable and renewable power source as energy recovery from MFCs improves, and this system will concurrently clean wastewater to provide purified water suited for diverse onsite uses. However, the most important obstacles preventing such systems from deployment worldwide are consumer perceptions of cleanliness and the reuse of human waste.

## 10. Challenges

In theory, MFCs may generate anodic half-cell potentials of 0.3 V and cathodic half-cell potentials of 0.8 V while employing oxygen as the cathodic terminal electron acceptor to achieve an overall maximum voltage of 1.1 V. However, due to polarization losses such as Ohmic, activation, concentration, and bacterial metabolism, the voltage produced from an MFC is too low and can be up to 0.8 V in open circuit and 0.6 to 0.7 V under external load situations. Even with innovative and effective cathodic catalysts, reducing cathodic overpotential remains difficult. The low voltage generated is insufficient to power electronic gadgets or function as an energy source. A number of MFC units must be joined in a stacked configuration to make this technology a viable option, resulting in voltage reversal in MFCs. Voltage reversal occurred in series-connected MFCs due to nonspontaneous anodic overpotential in some MFCs with delayed anodic reaction kinetics due to mass diffusion constraints.

## 11. Strategies for Enhanced Efficiency

Scaling up MFC technology has been hampered by various microbiological, electrochemical, engineering, economic, material science, and technical restrictions [22,90]. However, due to the poor efficiency of existing traditional site sanitation procedures, field deployment of this bioelectrochemical system in the field of sanitation can be realistic and practicable only by overcoming these scaling up limits. Moving in this approach, Kretzschmar et al. introduced the concept of the eLatrine MFC, employing low-cost corrugated cardboard electrodes for exploiting human feces as a substrate for microbial oxidation [23]. Du et al. employed a 1 L capacity MFC to treat human feces in 2009 [22]. The elimination of TCOD and ammonium was 71% and 44% in a dual-chambered MFC that ran for 190 h with a maximum power production of 70.8 mW/m^2^. In another study, an MFC fed with human feces wastewater after fermentation pre-treatment had a power density of 22 mW/m^2^, which was 47 percent greater than the control MFC that did not receive fermentation pre-treatment [91]. Lin et al. developed a novel idea for a microbial electrochemical septic tank (MEST), a modified configuration of CST, to improve traditional systems’ performance with minimum alterations and energy generation. In a 1 L capacity MEST operating at 25 °C, total phosphorous removal of 98.7 percent was accomplished, as was full sulfide removal [92]. Perlow previously employed human excrement as a substrate for a field-scale MFC project in Uganda in 2012 [93]. This unit’s power output was sufficient to cause a little light to blink. This MFC design cost roughly USD 20 and proved an effective MFC connection with a septic tank system; however, the low power output and related capital cost rendered it infeasible for practical use [83,94]. Different scaling up approaches are listed in Table 3.

## 12. Conclusions

Massive amounts of human waste are produced due to rapid urbanization and population expansion. Septic tanks are commonly used in underdeveloped nations for excreta storage and treatment. Septic tanks create human waste, which causes pollution and disease outbreaks. Alternatively, this waste may be used as a raw material for beneficial products, aiding in protecting our precious environment and human resources by limiting the spread of excreta-related illnesses. A choice matrix for the primary and sludge treatment options was created, considering the land need, energy demand, groundwater level, capital and operating costs, the skill required, and discharge standard. Using human feces as a substrate in n MFC can decompose waste pollutants while recovering electrical power. Integrating the MFC-centered toilet system into the regular sanitation behaviors of the user community is critical for the effective adoption of MFC as a sanitation technology. The successful trials of the bioelectric toilet system (India) and the Pee-power urinal (UK) are important subsequent indications demonstrating the MFC’s practical relevance to onsite sanitation. This paper described the deployment of a bioelectrochemical system into a septic tank to improve the treatment efficiency of existing sanitation methods and develops a loop for commercializing such a system. Such a sustainable and renewable MFC system may produce onsite power from human waste in remote areas and enhance existing sanitation practices, providing access to clean sanitation for the underprivileged class of emerging nations and the global population.

## Figures and Tables

**Figure 1 biotech-11-00036-f001:**
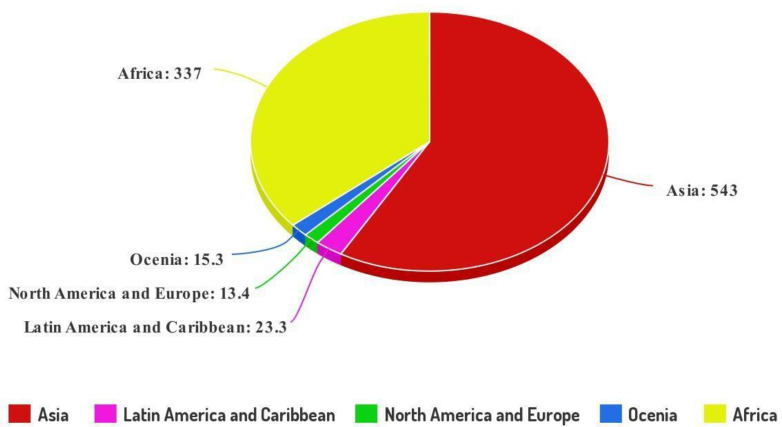
Global distribution of people (numbers in millions by region) without access to improved sanitation facilities in 2020. Source: WHO/UNICEF, 2020 [2].

**Figure 2 biotech-11-00036-f002:**
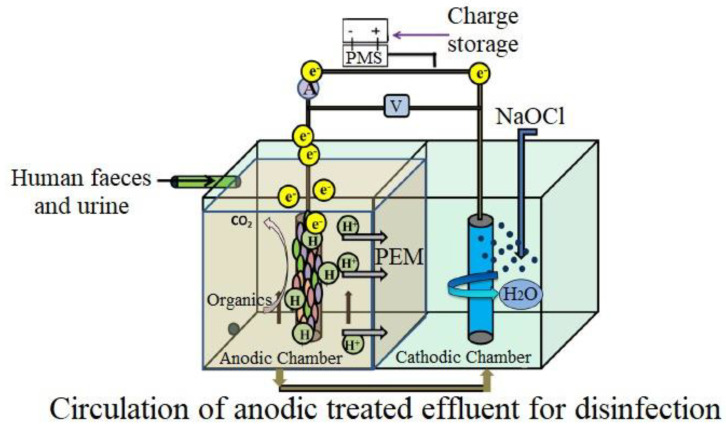
Schematic of human waste-processing microbial fuel cell. Anode chamber is fed with human wastes where different compounds are oxidized by catalytic action of bacteria. In cathode chamber, half-cell reduction takes place.

**Figure 3 biotech-11-00036-f003:**
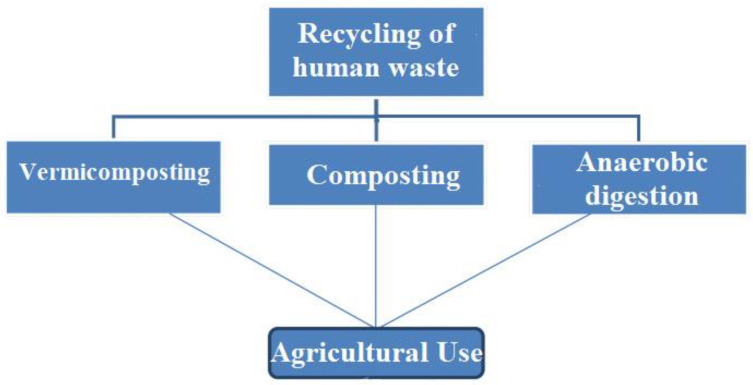
Conventional techniques for human waste treatment.

**Figure 4 biotech-11-00036-f004:**
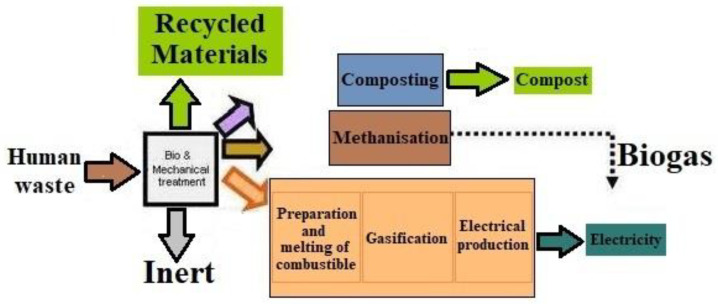
Modern methods for human waste treatment.

**Figure 5 biotech-11-00036-f005:**
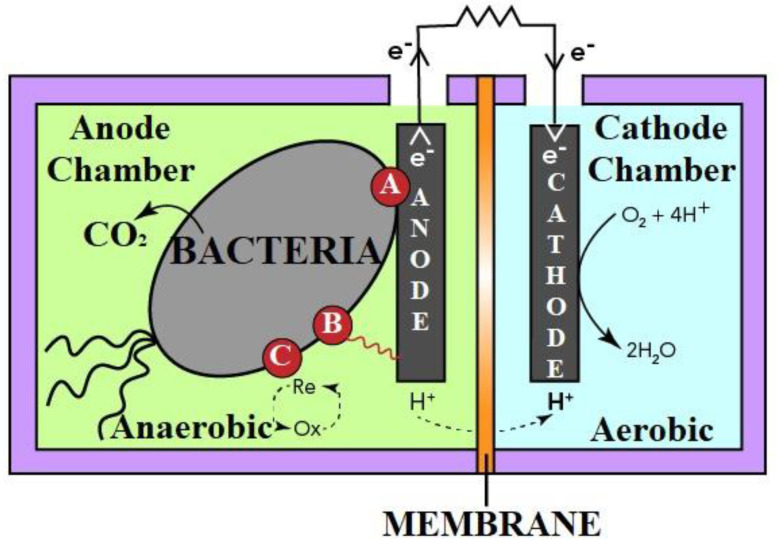
Schematic diagram of an MFC. The anode and cathode chambers are separated by a selective membrane for selective passage of protons.

**Figure 6 biotech-11-00036-f006:**
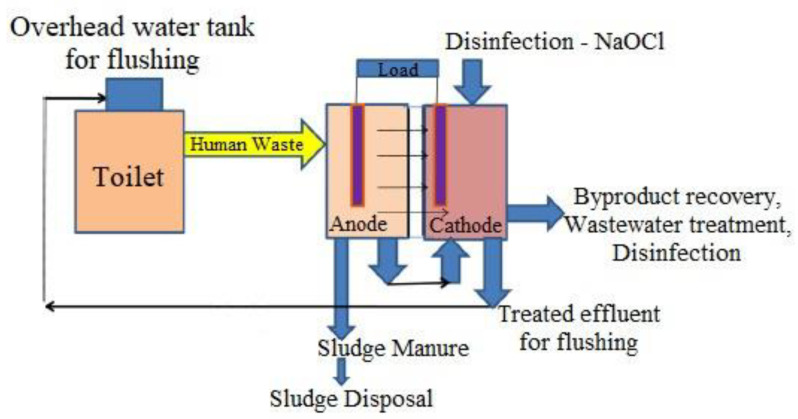
Flow chart depicting the efficient treatment of human waste in MFC.

**Figure 7 biotech-11-00036-f007:**
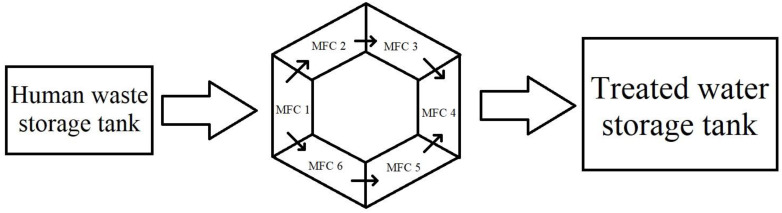
MFC arrangement of Sanitary Wastewater Treatment System. (adapted from Das et al. (2020) [82]).

**Figure 8 biotech-11-00036-f008:**
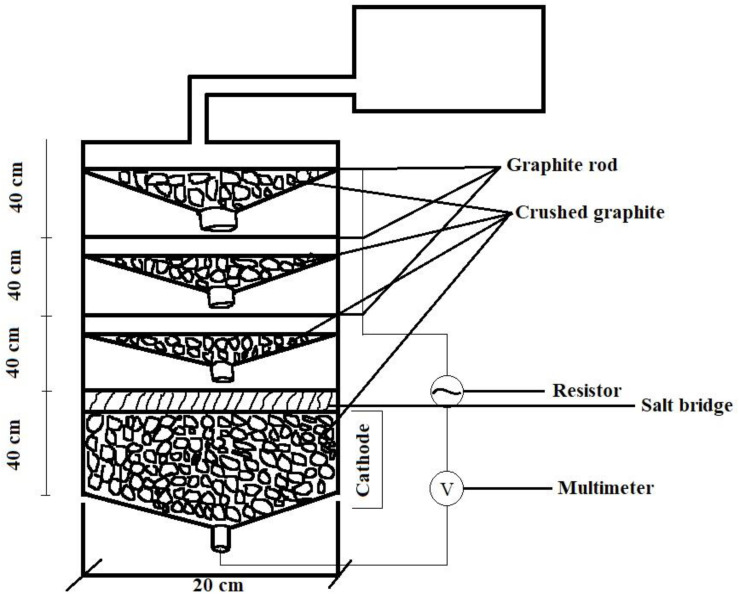
MFC arrangement of septic wastewater treatment (adapted from Leton et al. (2019) [68]).

**Figure 9 biotech-11-00036-f009:**
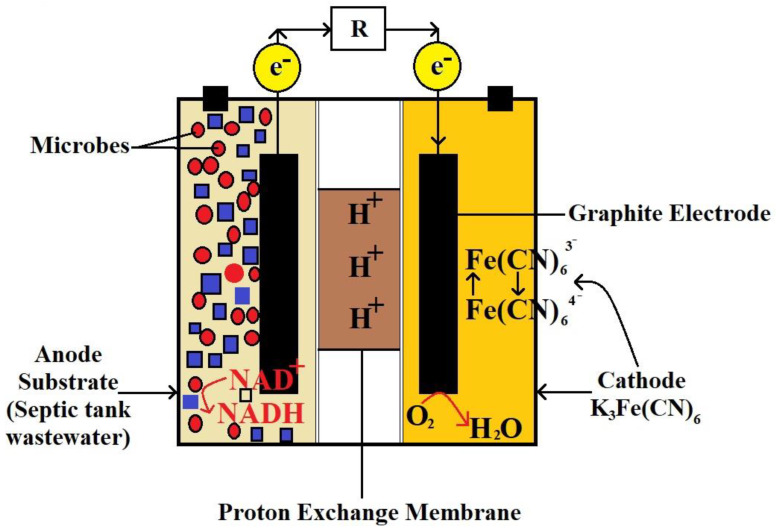
MFC arrangement of septic tank wastewater treatment (adapted from Thulasinathan et al. (2019) [69]).

**Figure 10 biotech-11-00036-f010:**
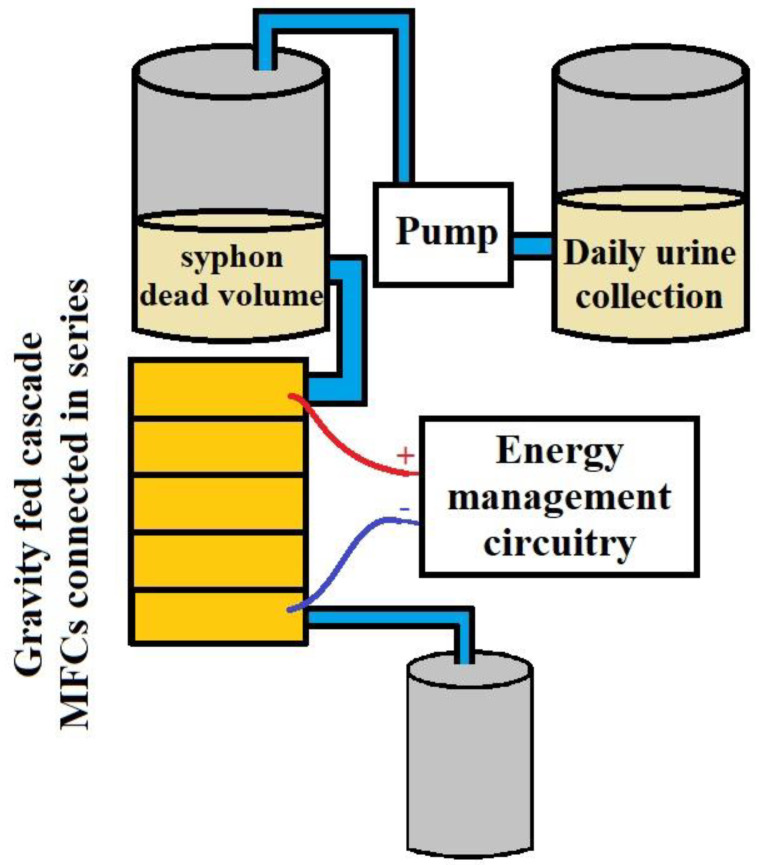
MFC arrangement of urine treatment and power generation (adapted from Walter et al. (2017) [83]).

**Table 1 biotech-11-00036-t001:** Comparison of various human waste management techniques.

S. No.	Technique	Working	Disadvantages
**1.**	Incineration Technology	Thermal treatment technology reduces the volume of waste requiring final disposal. Incineration can typically reduce the waste volume by over 90%.	Pollution during the incineration process is a potential risk to human health, and living or working near an incineration facility can have social, economic, and psychological effects.
**2.**	Autoclaving	Autoclaving is typically used for healthcare or industrial applications. An autoclave is a machine that uses steam under pressure to kill harmful bacteria, viruses, fungi, and spores on items placed inside a pressure vessel.	Use only limited quantities of waste.
**3.**	Microwaving	An alternative technology to the incinerator is a steam-based process and electromagnetic waves with frequencies between radio and infrared waves that use moisture inside the waste or additional steam to sterilize waste and destroy infectious agents and pathogenic organisms in the waste.	Not as effective as other sterilizing methods at killing bacteria; heat tends to be lower.
**4.**	Landfilling	The landfill is permanently capped with a plastic liner when it is full. After it is capped, the landfill is covered with two feet of soil. Then, vegetation (normally grass and plants without penetrating roots) is planted on top to prevent soil erosion due to rainfall and wind.	It can increase human health risks and environmental pollution if not handled carefully and properly.
**5.**	Plasma Pyrolysis	High temperature is produced using a plasma torch in an oxygen-starved environment to convert waste efficiently and in an eco-friendly manner.	High operation cost, large initial investment and low net energy production are some of its bottlenecks.

**Table 2 biotech-11-00036-t002:** Power density of a microbial fuel cell with varied substrates, substrate concentrations and resistance.

Source (Substrate)	Concentration of Feed (kg COD/m^3^)	Resistance (Ω)	Power Density (mW/m^2^)	Reference
**Cellulose**	0.4	10	0.02	[60]
**Phenol**	0.004	15	0.1	[61]
**Domestic Wastewater**	0.006	50	0.06	[62]
**Swine Wastewater**	0.009	10	0.7	[63]
**Urine**	0.25	10	4.508	[64]
**Human feces**	1	50	2.4	[22]
**Cow urine**	3	50	5.23	[14]

**Table 3 biotech-11-00036-t003:** MFCs used in scaling-up investigations for treatment of human waste and power generation.

Reactor Details	Volume (L)	Design Aspects	Power (mA)	COD Removal (%)	Reference
Single chamber MFC	0.13	Pt based catalyst	0.23	75	[95]
MPC stack of 24 MFCs	0.0063	Cathode with a microporous layer	1–1.2	-	[86]
Modular MFCs of 432 units	25	Field testing of Pee power urinals	800	95	[96]
Air cathode, Nafion PEM	2	15 cartridges of MFCs	124	89.67	[97]
Pluggable flow MFC	3	Column air-cathode MFC	142	-	[65]
Bioelectric toilet MFC	100	36 stacked MEAs	36	91	[98]
Hexagonal MFC	720	6 chambers	247	-	[82]
Multistage cylindrical MFC	20	5 sections	-	86.4	[68]
4 chambered concrete MFC	648	4 chambers	3	94	[99]
Bioelectric toilet	1500	6 chambers	239	95	[100]

## Data Availability

Not applicable.

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
