# Peer review of "Integrating Human Waste with Microbial Fuel Cells to Elevate the Production of Bioelectricity"

_biotech, 2022, doi:10.3390/biotech11030036_

Round 1
Reviewer 1 Report
In this review article, the authors summarize the utilization of human waste for electricity. The overall flow of the work is good and, in my opinion, a minor revision is needed before publication.
1. A typical review article should contain 4 parts: Introduction, Background knowledge, Examples, and Conclusion. However, this manuscript has 12 sections. Please combine some of them.
2. The first paragraph of the Introduction illustrates sanitation issues all over the world, which is a little irrelevant to the scientific topic of the manuscript. Please make this paragraph more concise.
3. Please provide references for Sections 2.1 and 2.2.
4. Table 1 needs to be re-formatted. It is hard to read.
5. Please provide references for Sections 4.1 and 4.2.
6. Please provide more figures from the literature that demonstrate current MFC studies.
Author Response
- A typical review article should contain 4 parts: Introduction, Background knowledge, Examples, and Conclusion. However, this manuscript has 12 sections. Please combine some of them.
Response: This article highlights the issue of human waste—which has great potential and could one day be used as a source of energy. This document is not a standard review because the authors put their utmost efforts to make it more helpful and knowledful through various techniques. This document has several particulars that require discussion. As a result, the authors request the editorial group and the reviewer not to merge the sections because doing so will cause this manuscript to lose its significance. Changes are made as per the reviewer’s recommendation (annotated version).
- The first paragraph of the Introduction illustrates sanitation issues all over the world, which is a little irrelevant to the scientific topic of the manuscript. Please make this paragraph more concise.
Response: For ease of comprehension, the text has been condensed (Line 44-46 , and 50-53, page 1-2).
- Please provide references for Sections 2.1 and 2.2.
Response: References have been provided (Line 92-172, page 3-5).
- Table 1 needs to be re-formatted. It is hard to read.
Response: Table 1 has been re-formatted (Line 175-176 page5-6).
- Please provide references for Sections 4.1 and 4.2.
Response: References have been provided (Line 229-239, page 8 and Line 240-250, page 8).
- Please provide more figures from the literature that demonstrate current MFC studies.
Response: Figures have been added (Line 403-414, page 12-14).

Reviewer 2 Report
1. Development of microbial fuel cells (MFC) integrating human waste for production of bioelectricity is reviewed in this article. Although the content of the review manuscript might provide helpful references for academicians in relevant fields, the manuscript is not well prepared. Major revision is thus requested prior to its being further consideration for possible publication in the journal.
2. At the end of Introduction on page 3, relevant study or review articles related to MFC integrating human waste could be further supplemented to emphasize the reviewing gaps of previous articles or possible contribution of the present review article.
3. Are references required to be cited for the figures in Figs. 2, 3, 4, 5, and 6?
4. Whole words could be provided and explained meanings once their corresponding abbreviations first appear in the text, for example SDG 6 and SDGs at line 125, ORR at line 171, CNTs at line 301, BET at line 323, etc.
5. At lines 246-247, what is the probable mechanism or reaction equation for energy release from the oxidation of organic molecules?
6. The sentences at lines 258-263 are repeated with those at lines 247-253.
7. Adequate references could be cited for a few statements for example at lines 244-261 and at line 249 after “fermentation”.
8. Typo errors appear, for example: at lines 296-297, “cm2”; at line 343, “142 6.71”, at line 356, “0.45 0.65V”,
9. Wrongly cited references appear, for example Ref. [52] are not authored by Kretzschmar at line 522; no reference is cited for Du et al. at line 524; at lines 530-532, “Lin et al.” not corresponding to Ref. [62-63], etc. Hence, the references in this article could be carefully checked again.
10. The last two paragraphs in Conclusion at lines 563-569 are redundant and should be removed.
Author Response
Reviewer #2:
- Development of microbial fuel cells (MFC) integrating human waste for production of bioelectricity is reviewed in this article. Although the content of the review manuscript might provide helpful references for academicians in relevant fields, the manuscript is not well prepared. Major revision is thus requested prior to its being further consideration for possible publication in the journal.
Response: We thank the reviewer for the valuable comments and suggestions. The manuscript has been revised extensively incoorporationg new references and figures.
- At the end of Introduction on page 3, relevant study or review articles related to MFC integrating human waste could be further supplemented to emphasize the reviewing gaps of previous articles or possible contribution of the present review article.
Response: Relevant material has been added.
- Are references required to be cited for the figures in Figs. 2, 3, 4, 5, and 6?
Response: No, all figures are created entirely by the authors, so no citation is required.
- Whole words could be provided and explained meanings once their corresponding abbreviations first appear in the text, for example SDG 6 and SDGs at line 125, ORR at line 171, CNTs at line 301, BET at line 323, etc.
Response: Meaning for abbreviation has been provided for Line 120-121, page 4; Line 169-170, page 5; Line 298, page 9 and Line 320, page 9.
- At lines 246-247, what is the probable mechanism or reaction equation for energy release from the oxidation of organic molecules?
Response: The reduction reaction from the organic oxidation is conversion og oxygen into water, as shown in equation under section 3.
- The sentences on lines 258-263 are repeated with those at lines 247-253.
Response: Repeated lines are deleted.
- Adequate references could be cited for a few statements for example at lines 244-261 and at line 249 after “fermentation”.
Response: References have been provided (Line 240-260, page 8).
- Typo errors appear, for example: at lines 296-297, “cm2”; at line 343, “142 6.71”, at line 356, “0.45 0.65V”,
Response: Error has been fixed at Line 285-286, page 9; Line 332, page 10 and Line 345, page 11.
- Wrongly cited references appear, for example Ref. [52] are not authored by Kretzschmar at line 522; no reference is cited for Du et al. at line 524; at lines 530-532, “Lin et al.” not corresponding to Ref. [62-63], etc. Hence, the references in this article could be carefully checked again.
Response: The incorrectly cited reference has been corrected (Line 604-606, page 6; Line 606-607, page 16 and Line 612-616, page 16-17).
- The last two paragraphs in Conclusion at lines 563-569 are redundant and should be removed.
Response: Paragraphs have been removed.

Round 2
Reviewer 2 Report
The authors have replied most of the reviewer’s comments. However, some comment has not responded or explained yet, as stated below:
(1) In response to the previous comment No. 2, while the authors only replied with “Relevant material has been added”, please respond more clearly what and where the added text are in the revised manuscript.
(2) Please move the title of Table 2 at lines 317-318 to the top of this table.
Author Response
- In response to the previous comment No. 2, while the authors only replied with “Relevant material has been added”, please respond more clearly what and where the added text are in the revised manuscript.
Previous comment: At the end of Introduction on page 3, relevant study or review articles related to MFC integrating human waste could be further supplemented to emphasize the reviewing gaps of previous articles or possible contribution of the present review article.
Response: Authors are deeply regretted that this comment was not satisfactorily addressed in the previous version. Now we have included a few relevant studies related to MFC integrating human waste and cited relevant articles.
- Please move the title of Table 2 at lines 317-318 to the top of this table.
Response: Table 2 has been fixed, “line 356-358 page 10-11”. Changes are made as per reviewer’s recommendation (annotated version).
